



# Characteristics and performance of vertical winds as observed by the radar wind profiler network of China

Boming Liu,[1,+] Jianping Guo,[2,+] Wei Gong,[1] Lijuan Shi[3], Yong Zhang[3] and Yingying Ma[1]

[1] State Key Laboratory of Information Engineering in Surveying, Mapping and Remote Sensing (LIESMARS), Wuhan
  University, Wuhan, China
[2] The State Key Laboratory of Severe Weather, Chinese Academy of Meteorological Sciences, Beijing 100081, China
[3] Meteorological observation Centre, Chinese Meteorological Administration, Beijing 100081, China
[+] The authors contributed equally to this work

*Correspondence to*: Jianping Guo (jpguocams@gmail.com)

**Abstract.** Vertical wind profiles are the foundation for numerical weather prediction systems research. Large-scale vertical wind data have been previously documented from network observations in several countries, but the nationwide vertical wind observations are poorly understood in China. In this study, the salient characteristics and performance of vertical winds as observed by the radar wind profiler network of China was investigated, which consists of more than 100 stations instrumented with 1290-MHz Doppler radar designed primarily for measuring vertical-resolved winds. This network has good spatial coverage, with denser sites in coastal areas. The vertical wind profiles observed by this network can provide the horizontal wind direction, horizontal wind speed, and vertical wind speed for every 120 m interval within the height of 0 to 3 km. The availability of the radar wind profiler network has been investigated in terms of effective detection height, data acquisition rate, data confidence, and data accuracy. Further comparison analysis with reanalysis data indicated that the observation data at 89 stations are recommended, and 17 stations are unrecommended. The vertical wind profiles can serve as important input dataset assimilated into numerical weather prediction systems at both regional and global scales.



# 1 Introduction

It is increasingly recognized that the atmospheric vertical wind profile and its vertical wind shear are crucial to better understanding the more frequent extreme rainfall events (Huuskonen et al., 2014; Nash et al., 2001; Weber et al., 1990), intensification of clear-air turbulence associated with aircraft

safety (Williams & Joshi, 2013), complicated aerosol-cloud-precipitation interaction (Fan et al. 2009; Guo et al. 2016a; 2019; Lee et al., 2016), and persistent particulate pollution episodes (Yang et al., 2019; Zhang et al., 2020). For the wind speed in the planetary boundary layer (PBL), the most striking feature is that the turning of winds with height dominates the whole PBL and beyond, which can be explained in terms of force vectors (drag, pressure gradient force, Coriolis force) at the

surface and the top of the ABL (pressure gradient force and Coriolis force) (Lemone et al., 2018). Under influences of large-scale dynamic forcing and land surface process, wind speed and direction will dramatically vary (Michelson & Bao, 2008), which poses a large challenge for models to simulate or forecast the variation of wind very well, especially in the PBL (Constantinescu et al., 2009; Guo et al., 2016b; Liu et al., 2017; Yim et al.., 2007).

Radar wind profiler (RWP), which is generally Doppler radar that operates at either the VHF (30-300 MHz) or UHF (300-1000 MHz) frequency bands, has been widely applied to atmospheric vertical wind field research (Dolman et al., 2018; Molod et al., 2015; Ishihara et al., 2006; Nash et al., 2001; Schlatter et al., 1994). To date, a large spectrum of field campaigns involved with the RWP derived vertical winds, especially over the regions with intensive anthropogenic and industrialized activities,

have been conducted and their archived dataset has been increasingly received much attention (Liu et al., 2019; Kottayil et al., 2016; Singh et al., 2016; LeMone et al., 2013; Bianco et al. 2008; Le et al., 1998), most of which are based on ground-based remote sensed measurements. Since 2018, satellite-based wind observational era set in with the launch of European space agency (ESA)'s Aeolus wind satellite on which the direct-detection Doppler wind lidar ALADIN is accommodated (Reitebuch et

al., 2009; Reitebuch 2012), despite its dataset still not released for the public access and use.



Moreover, aircraft-based observation of atmospheric vertical wind has been carried out worldwide (Lux et al., 2018; Marksteiner et al., 2018; Witschas et al., 2017; Chouza et al., 2016; Weissmann et al., 2005). Especially in the North Atlantic Waveguide and Downstream Experiment (NAWDEX) campaign, the ALADIN team conducted several airborne wind measurement experiments for the validation of the Aeolus satellite winds product (Lux et al., 2020; Zhai et al., 2020).

To gain a full picture of vertical wind over regional scale, and to fill the data gap left by field campaign, a number of RWP networks have been set up across the world. As early as 1990s, the demonstration wind profile network was deployed and maintained by the National Oceanic Atmospheric Administration (NOAA), which was also termed NOAA profiler network (NPN) and operated at a frequency of 404 MHz (Schlatter et al., 1994; Vande Kamp, 1993; Weber at al., 1990). The second type of profiler was the 915-MHz boundary-layer profiler that was much smaller, transportable, commercially available but lacked height coverage compared with 404 MHz wind profiler, and thus was mainly used for NOAA research and outside agencies. To make the most of the best sampling attributes of the abovementioned two types of wind profiler, a third type of profiler operated at 449 MHz. Later on (in 1996), the European Cooperation in Science and Technology framework (COST) initiated the project of Wind Initiative Network Demonstration in Europe (CWINDE). Under the framework of CWINDE, the European RWP network named E-PROFILE, as part of the EUMETNET Composite Observing System (EUCOS), was constructed, providing the monitoring of vertical profiles of wind across Europe (Dibbern et al., 2001; Oakley et al., 2000; Nash et al., 2000). Moreover, the Japan Meteorological Agency developed the operational wind profiler network in Japan in 2011, which was a nationwide network of 33 wind profiler currently in operation. The wind data have significant positive impact on improving numerical weather prediction (Ishihara et al., 2006). The Australian Government Bureau of Meteorology completed the installation of the Australian wind profiler network of 19 wind profiler in 2017 that runs at 55 MHz frequency band, which produced wind data



of sufficient accuracy for presentation to forecasters and ingestion into global numerical weather prediction models (Dolman et al., 2018). The aforementioned networks have provided the vertical profiles of wind for model assimilation through the Global Telecommunication System at regional or national scale (e.g., Benjamin et al., 2004; Chipilski et al., 2019), which improved the forecast of the rainfall onset and the atmospheric pollutants (Liu et al., 2019; Liu et al., 2018; Singh et al., 2016; LeMone et al., 2013; Du et al., 2012; Bianco et al. 2008; Angevine et al., 1994).

Given the considerable advantages over conventional ground-based in situ or remote sensing observations, wind profiler measurements have been well applied in a variety of applications in China, including air quality and weather forecast (Sun, 1994; Hu et al., 2010; Dong et al., 2011; Miao et al. 2018; Zhang et al., 2020). Nevertheless, the RWP was generally deployed at either specific regions or short time period. Recent model simulation work by assimilating wind measurements from a regional wind profiler network in North China indicated the network observation significantly improved the convective forecasting (Wang et al., 2020). Meanwhile, the extreme precipitation is continuously intensified under global warming and deteriorated atmospheric pollution, especially in Eastern Asian countries such as China and India (Zhang et al., 2006; Pfahl et al., 2017; Guo et al., 2019; Li et al., 2020), which desperately needs the vertical wind observations. However, the characteristics and performance of nationwide profiler network in China has never be revealed, and the assessment of systematic observation performance and data accuracy are still lacking, to the best of our knowledge. This motivates us to evaluate the performance and accuracy of radar wind profiler network of China, ultimately in an attempt to present a vertical wind profile data as a new reference for numerical weather prediction or climate related studies. The remainder of this paper is organized as follow. The radar wind profiler network of China are briefly introduced in Section 2. The performance and accuracy are evaluated in Section 3. Section 4 will discuss the detailed application of vertical wind profile data. A summary of results is presented in Section 5.



## 2 Description of the radar wind profiler network

The radar wind profiler network of China is comprised of 106 stations until March 2019, which is designed primarily for measuring winds at various altitudes. This network began to be constructed dating back to 2008, when there were 5 sites having wind profiling measurements transmitted to the

headquarter of China Meteorological Administration (CMA). The number of RWP sites continuously increased to 92 at the end of 2017, all of which were operating at 405 MHz frequency band. Afterwards, the working frequency band was changed to L band (1290 MHz), and the number increased to 128 at February 2020 (personal communication with Dr. Ruiyi Li from CMA). The Meteorological Observation Center (MOC) of CMA is responsible for the operation and maintenance

of the nationwide wind profiler network. The majority of the radars operate at L band (101 sites), and a few of the radars operate at P band (5 sites). Figure 1 shows the spatial distribution of wind profiler network in China, which exhibits large spatial domain extending from the northernmost site located at Wulumuqi at 43.0°N, to the southernmost one at Nanhai at 17.0°N, and from the westernmost site also located at Wulumuqi at 87.0°E to the easternmost one in Shenyang at 123.0°E.

The MOC/CMA was responsible for the maintenance and collection of wind measurements from the wind profiler network, as shown in Fig. 2. Specifically, the data transfer from radar sites to MOC/CMA was mainly done using Internet connections. The data center of CMA was established to efficiently process the data collected by Internet. There are two main types of data collected from the wind profilerr network: one is raw data and the other product data. The former data includes the

power spectrum data files (indicated by FFT) and radial data files (indicated by RAD). The power spectrum data file is composed of file identification, basic parameters of the station, performance parameters, observation parameters and observation data. The power spectrum data file is dynamically generated in real time according to one's demand. The radial data files are twofold: one is reference information, such as the basic parameters of the station, radar performance parameters,

and observation parameters; the other is the observation data of each beam at each sampling height,



including sample height, velocity spectrum width, signal-to-noise ratio, and radial velocity. As for the product data, three main wind profile products are produced operationally by the data center of CMA: (1) Real-time sampling data file (at 6-min intervals), mainly including the sampling height, horizontal wind direction, horizontal wind speed, vertical wind speed, horizontal credibility, vertical

credibility, and Refractive Index Structure Parameter ($C_n^2$). An individual file will be produced for every 6-min detection and is marked as ROBS. (2) Half-hour data file (at 30-min intervals), which is broadly consistent with ROBS file in terms of both data content and format, except for the file produced for every half hour (48 files per day), and the file is marked as HOBS. (3) One-hour observation sampling data file (at 60-min intervals) with 24 files per day, which is marked as OOBS.

These wind profile products are generated on each observation sites. The vertical resolution of wind profile data at most sites is 120 m. However, in a few sites, due to the initial parameter was set to low detection mode and high sampling rate, the vertical resolution of wind profile data is 60 m. Examples of vertical wind profile product are shown in the Fig. 3. Seven different heights (150, 500, 1000, 1500, 2000, 2500, and 3000 m) selected to show the atmospheric vertical wind field. It can provide

the vertical profile of horizontal wind direction, horizontal wind speed, and vertical wind speed. Those products are available for official duty use and for research and education. The observation data from November 2018 to March 2019 were used to evaluate the performance of the radar wind profiler network of China.

## 3 Performance of the radar wind profiler network

The radar wind profiler network of China includes a variety of types of wind profile radars, including tropospheric type I, tropospheric type II and boundary layer wind profile radars. Due to the algorithms and setting parameters of different instruments are inconsistent, the system performance index and data accuracy are inhomogeneity. Therefore, it is necessary to evaluate the system





performance index and data accuracy of the radars in the radar wind profiler network. This was a major step forward in the harmonization of the product generation and data quality of the radar wind profiler network of China. Three system performance indicators on data application were investigated, including effective detection height, data acquisition rate, and data confidence.

Moreover, the vertical wind profile data from RWP was also compared with ECMWF wind data to verify the accuracy of the data.

*3.1 System performance index*

The operation mode of the radar wind profiler network includes high, medium, and low detection modes, which can detect wind field information at different altitudes. High mode was generally used

to detect the wind fields at a height of 5 ~ 10 km above ground level (AGL). The medium and low modes was used to detect wind fields at a height of 0 ~ 5 km above the ground. We here define "effective detection height" as the effective detection height up to where wind measurements are available. Figure 4a, b shows the mean effective height detected by each RWP during the period from November 2018 to March 2019. There are 90 stations with an average height greater than 3 km;

10 of them can even reach more than 7 km. As for the acquisition rate, it refers to the ratio of the actual acquisition time to the total theoretical acquisition time, which is used to evaluate the normal operation of the wind profile radar. Figure 4c, d represents the data acquisition rate of wind measurement at radar wind profiler network during the period from November 2018 to March 2019. The data collection rate of most sites is greater than 90%, while the data collection rate of 4 sites is

less than 50%. Figure 4e, f represents the average confidence of wind measurement at radar wind profiler network. Confidence is a credible parameter set by the system for the wind speed information at each sampling point, which is used to evaluate the credibility of the wind field information retrieved at each altitude position. The results indicated that there are 100 sites with 100% confidence, but 6 sites have less than 100% confidence.



In order to unify the use criteria of radar wind profiler network data, we set corresponding screening criteria for each system index. The main purpose of the radar wind profiler network is to continuously monitor the atmospheric wind field near the ground (0-3 km). Therefore, the effective detection height needs to reach 3 km, and the acquisition rate must be above 60%. In addition,

according to the user manual of RWP, only wind profile data with a 100% confidence level are recommended. According to these criteria, the wind profile data at each site were screened, and the screening results are shown in Figure 5. Figure 5a shows the results of screening for effective detection height. The results show that the effective height detected by RWP of 102 stations meets this standard, and 4 stations are not up to the standard. The substandard sites are 54752, 58365,

58474, 58730. Figure 5b shows the screening results of the data acquisition rate. The results show that the data acquisition rate of 100 sites is satisfactory, and 6 sites are not up to standard. These substandard sites are 16078, 58158, 58460, 58927, 58933 and 59431. Figure 5c shows the results of the confidence level screening. The results show that 100 sites are up to standard and 6 sites are substandard. The substandard sites are 54727, 54736, 54857, 57494, 58365 and 58460. Overall, 92

sites of the radar wind profiler network have a good system performance.

*3.2 Data accuracy*

The echo signal from RWP can be processed to provide the vertical wind profile at RWP sites. However, it should be noted that the accuracy of wind profile data is also closely related to the inversion algorithm. Therefore, the work to verify the accuracy of the data is necessary before

applying these observation data. The comparison statistics against the wind profile data from ECMWF numerical model is an important monitoring tool (Huuskonen et al., 2014; Haseler, 2004). Figure 6 shows the comparison results between wind profile from RWP and that from ECMWF at six typical stations. The vertical validation range is from 0 to 3 km. The red and blue dot lines represent the mean bias and RMSE at different height, respectively. The vertical distribution of

deviation at different sites is different, but almost of deviations are less than 5 m/s. It is clear that a

discrepancy does not automatically imply that the wind profile is in error, but in general a gross deviation with the model results can be considered as an indication of a radar error. Ishihara et al. (2006) evaluated the wind accuracy of WINDAS by comparisons with the numerical weather prediction model profiles, and the RMSE were around 3 m/s. Huuskonen et al. (2014) compared the wind profiles observed by EUMETNET with the ECMWF model profiles, and set 5 m/s RMSE as a target for acceptable wind observations.

Here, we set 4 m/s mean absolute bias and 6 m/s RMSE as a target for acceptable criterion. Figure 7 shows mean absolute bias and mean RMSE from 0 to 3 km for all RWP, calculated by comparing with ECMWF wind data. It is seen that most of RWP meet consistently the acceptance criterion of 4 m/s mean absolute bias and 6 m/s RMSE, while few radars also show larger differences. Moreover, the data bias of radar wind profiler network has a certain spatial difference. According to the average bias in zonal direction (histogram in Fig.7), the RWPs at 28-32°N area have relatively large deviation, where the zonal mean absolute bias was larger than 2 m/s and zonal mean RMSE was larger than 5 m/s. The sites with mean absolute bias greater than 4 m/s include 54857, 57494 and 59046; and the sites with RMSE greater than 6 m/s include 52889, 57494, 58448 and 59046. The wind data at these sites have large deviations and are not recommended. The large difference may be caused by hardware problems or configuration problems but is mostly related to flaws in the algorithm. In addition, there are eleven wind profile radar sites which are equipped with radiosonde (51463, 54342, 54511, 54727, 54857, 57494, 57516, 58238, 59758, 59948, and 59981). Their performance can also be compared to the corresponding radiosonde, the data quality of the them is comparable when the RMSE between RWP and radiosonde was small.

Overall, the availability of radar wind profiler network of China can be evaluated by combining the system performance index and data accuracy. Figure 8 shows the spatial distribution and number of recommended and unrecommended sites. The availability of radar wind profiler network of China is 84%, which 89 stations are recommended, and 17 stations are unrecommended. These



unrecommended sites include: 16078, 52889, 54752, 54727, 54736, 54857, 57494, 58158, 58365, 58448, 58460, 58474, 58730, 58927, 58933, 59046, and 59431. For the sites with low height coverage or low data acquisition rate, the data availability can be improved by changing the radar observation modes and increasing radar runtime. But for the sites with low confidence level or low

data accuracy, which is caused by the inversion algorithm or the instrument system, it needs to choose the appropriate optimization method for specific problems. Some methods on data quality control are given in previous studies (Holleman, 2005).

## 4 Applications of the radar wind profiler network

### 4.1 Extreme weather detection

The vertical wind profile data can be used to monitor the extreme weather. Figure 9 presents the spatial distribution of diurnal phase and amplitude of wind speed averaged during the period from November 2018 to March 2019 according to mean maximum wind speed within the 24 h. To highlight the vertical detection capabilities of wind radar, mean maximum wind speed at four different heights above ground level (500, 1000, 1500 and 2500 m) were investigated. As shown in

the Fig. 9a (at 500 m), among the 106 observational sites, mean maximum wind speed occurs in the morning at 76 sites (about 71.9%), followed by 12 sites (11.3%) with peaks in the early morning. On the other hand, only 6 sites (5.5%) have the afternoon peak, whereas 12 sites (11.3%) have the evening peak. The story with respect to the diurnal phase and amplitude of mean maximum wind speed at other height is almost the same (Fig. 9b-d). In terms of vertical direction, the occurrence

timing of mean maximum wind speed at most stations is consistent; but some stations of northwest China (Wulumuqi, Lanzhou and Qinghai) is inconsistent. Moreover, the amplitude of mean maximum wind speed at 2500 m height is twice or three times than that at other height, indicative of the maximum wind speed increases with the height. In terms of the spatial pattern, mean maximum





wind speed generally occurs in the morning in the coastal region of eastern China, with magnitude generally lower than 10 m/s. By comparison, both early morning and afternoon peaks contribute almost equally to the diurnal cycle in the inland region.

*4.2 Regional wind field analysis*

The vertical wind profile data can also be used to investigate the regional wind field. As shown in Fig. 10, eleven regions of interest (ROI) selected, according to the spatial distribution of RWP stations. The site number and land cover type of each ROI was shown in Table1. The land cover types data in each ROI was obtained from MODIS. Figure 10 shows the atmospheric wind field variation of each ROI at 500 m above ground level during the study period. From the perspective of

wind direction, the North China Plain was mainly southwest wind during the study period, the southwest wind at ROI 3 and 4 accounted for 40.3% and 48% respectively. The south China area was mainly dominated by northeast wind, such as ROI 8, 9, 10, and 11. The distribution of wind direction over central China is more uniform. The western China is dominated by northwest wind, and the percentages of northwest wind at ROI 1 is 45.8%. In terms of the spatial pattern wind speed,

the wind speed in the western China is relatively low. The percentages of wind speed less than 4 m/s at ROI 1, 5, and 7 were 76.2%, 78.7%, and 83.2%, respectively. Moreover, the land cover type of ROI 1, 5, and 7 was both the grassland. By contrast, the wind speed in the central and eastern regions is significantly large, and 60% of the wind speed in most ROI can reach 6 m/s. Especially in coastal areas, such as ROI 4 and 9, the 30% of wind speed was larger than 8 m/s at the whole study period.

In the long run, the accumulation of more wind profile measurement across China, especially in the lowest part of PBL, will provide a valuable benchmark database for assessment of wind power potentials (Yim et al., 2007). The policymakers will determine whether the wind turbines will be installed or not, aided by high-resolution model simulation analyses. Moreover, the real-time wind field data can be used to predict typhoon and sandstorm paths (Ishihara et al., 2006; Huuskonen et al.,



2014). The radar wind profiler network of China can provide powerful data support for disaster warning and air pollution prevention.

## 5 Concluding remarks

The vertical wind profile is of great importance to the accuracy of numerical weather prediction

model, the prediction of precipitation, the diffusion of air pollution, research on regional climate changes, and site selection of wind power plants. To the best of our knowledge, we for the first time reported on the height-resolved winds starting from ground surface to as high as 3-10 km, based on the radar wind profiler network of China, which consisted of more than 100 RWP stations. It can provide the vertical profiles of horizontal wind direction, horizontal wind speed, and vertical wind

speed. Then, the availability of the radar wind profiler network was investigated from system performance index and data accuracy. The evaluation criteria are that the height coverage must reach 3 km, the data acquisition rate must exceed 60%, and the data confidence must be 100%. In addition, in terms of data accuracy, the mean absolute bias need to less than 4 m/s and RMSE less than 6 m/s. Under this criterion, the availability of the radar wind profiler network of China is 84%, which 89

stations are recommended, and 17 stations are unrecommended. Finally, the vertical wind profile data has a wide range of applications, such as extreme weather detection and regional atmospheric wind field research. This radar wind profiler network would serve as a key data source on spatiotemporal distribution of atmospheric wind field in support of related scientific researches and industrial applications in the future.

### Data availability

The Radar wind profiler data were provided by the National Meteorological Information Center of China Meteorological Administration at http://data.cma.cn/en/ (last access: 4 March 2020). The ECWMF dataset can be downloaded from https://cds.climate.copernicus.eu/ (last access: 24

February 2020). Instructions for use and data download methods can be found on the official website.



**Author contributions.**

The study was completed with close cooperation between all authors. J. Guo and B. Liu designed the idea for assessing the radar wind profiler data in China; J. Guo and B. Liu conducted the data analyses and co-wrote the manuscript; L., Shi, Y. Zhang, Y. Ma and W. Gong discussed the experimental results, and all coauthors helped reviewing the manuscript and the revisions.

**Competing interests.**

The authors declare that they have no conflict of interest.

**Acknowledgements.**

We are very grateful to the China Meteorological Administration for installment and maintenance of the radar wind profiler observational network. This work was financially supported by the National Key Research and Development Program of China under grants 2017YFC0212600 and 2017YFC1501401), the National Natural Science Foundation of China under grants 41771399, 41401498 and 41627804).

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





**Table 1.** Statistics of the number of sites and land cover types for the 11 regions of interest (ROI).

| Region of interest | Site number | Land cover types |
|---|---|---|
| 1 | 1 | Grassland |
| 2 | 7 | Cropland and Forest |
| 3 | 10 | Urban |
| 4 | 2 | Cropland |
| 5 | 2 | Grassland |
| 6 | 1 | Cropland |
| 7 | 1 | Grassland |
| 8 | 27 | Urban |
| 9 | 2 | Cropland and Forest |
| 10 | 10 | Urban |
| 11 | 19 | Urban and Forest |



# Figures

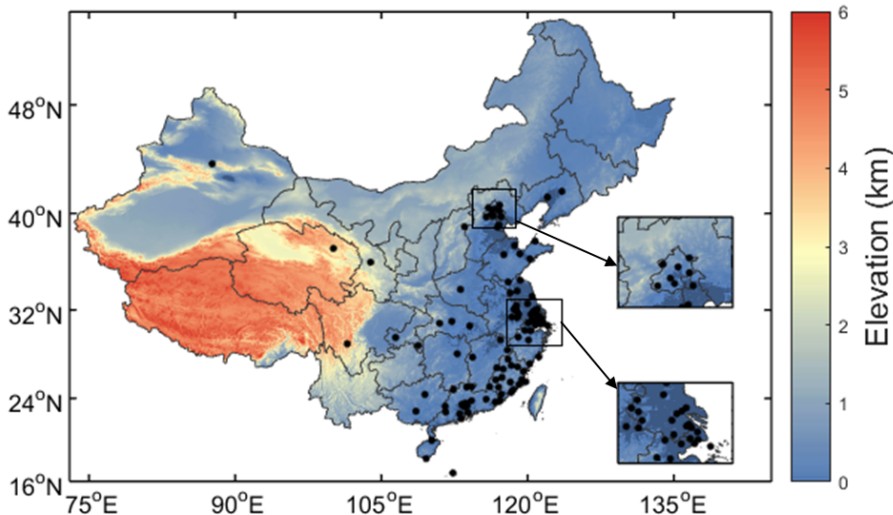

5 **Figure 1.** The site distribution of radar wind profiler network of China. Color bar means the elevation.





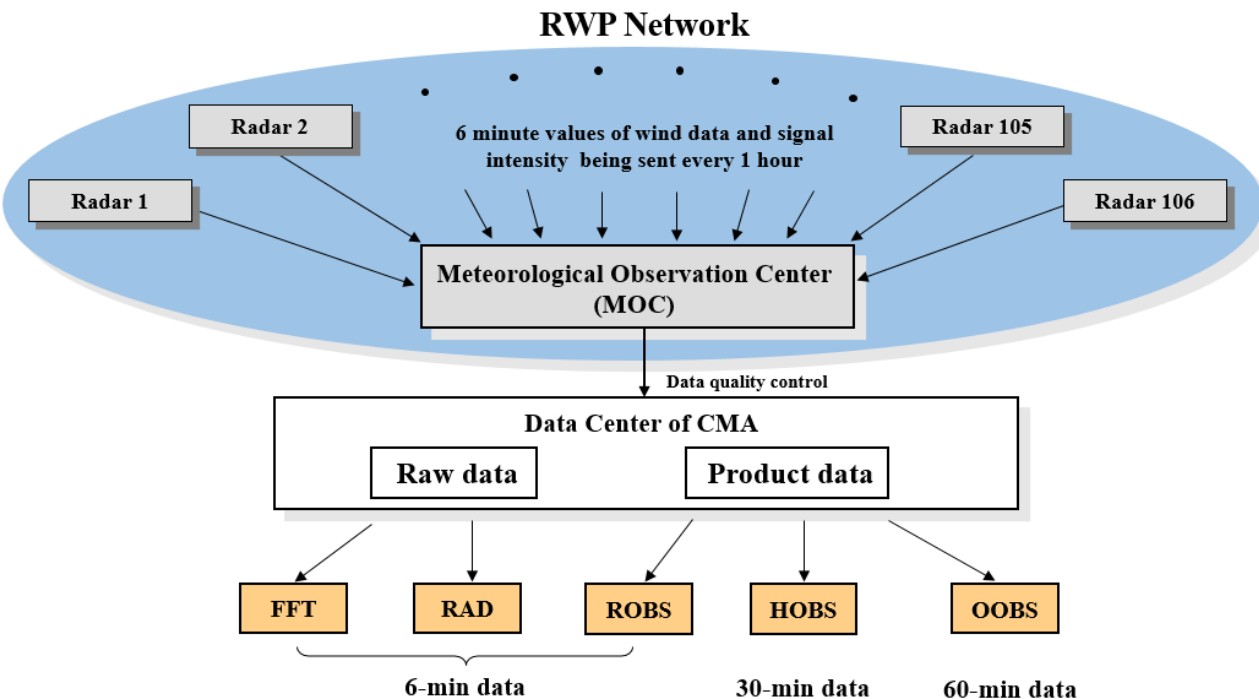

**Figure 2.** Data transmission framework of radar wind profiler network of China. The RWP network is operationally maintained by the Meteorological Observation Center (MOC), China Meteorological Administration (CMA).

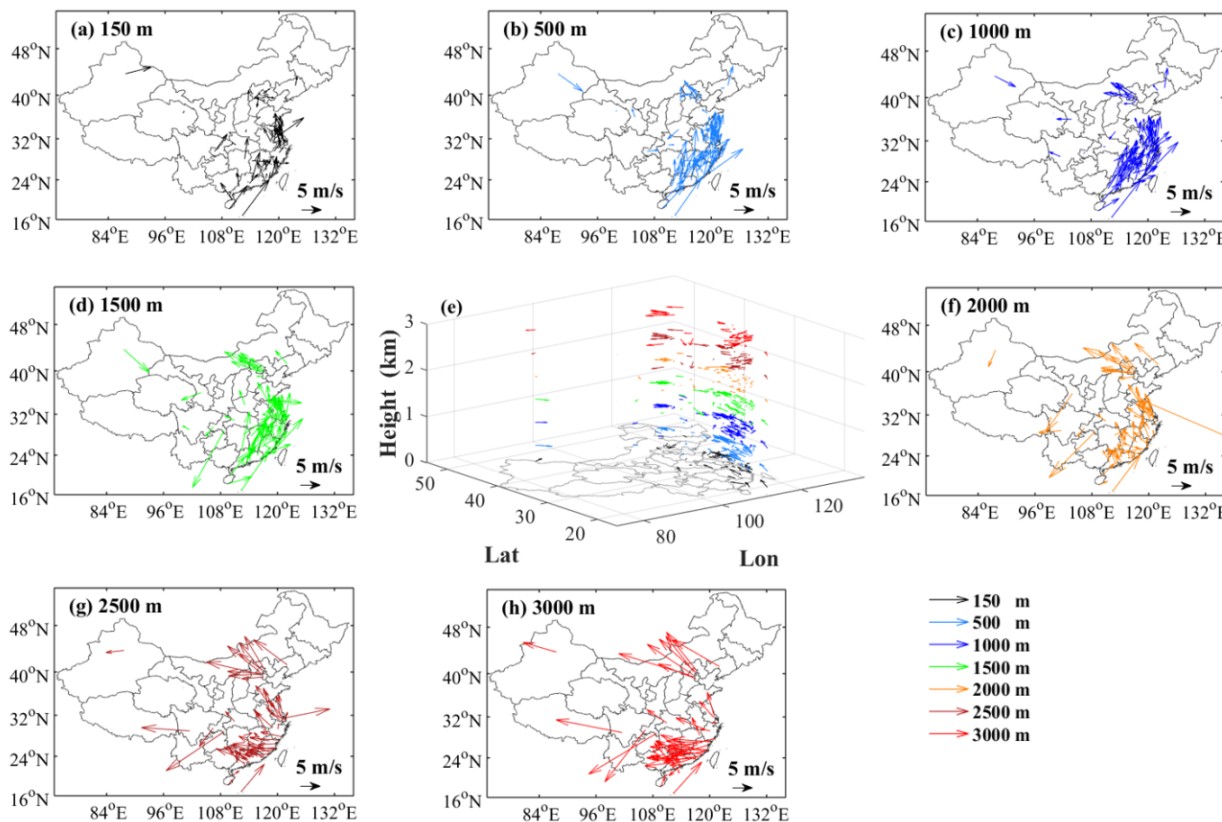

**Figure 3.** Spatial distribution of average wind field under different height: (a) 150 m, (b) 500 m, (c) 1000 m, (d) 1500 m, (f) 2000 m, (g) 2500 m, and (h) 3000 m above ground level (AGL). Also shown is (e) the three-dimensional atmospheric wind field observed by the operational radar wind profiler network of China.

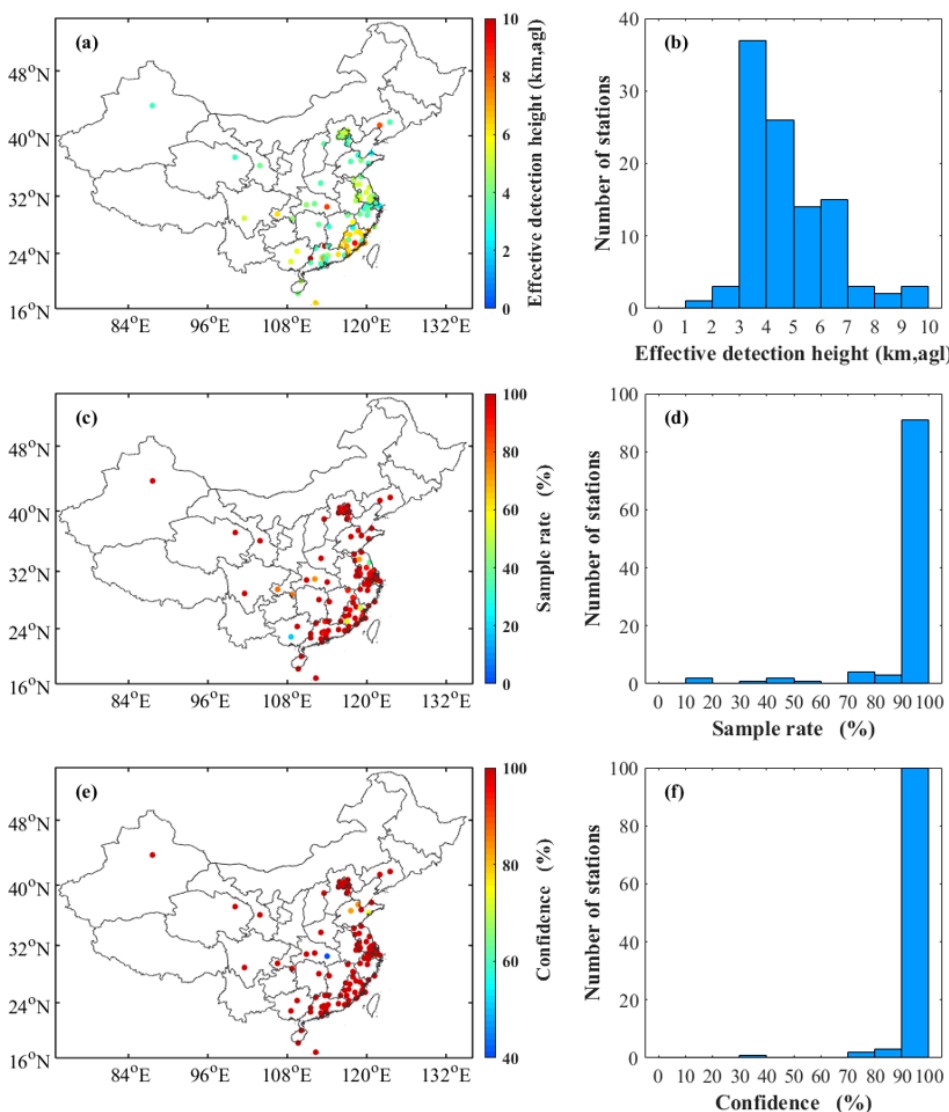

**Figure 4.** Spatial distribution of (a) mean effective detecion height, (c) mean data acquisition rate, and (e) mean data confidence at each station during November 2018 to March 2019; (b), (d), and (f) are corresponding histograms for (a), (c), and (e), respectively.





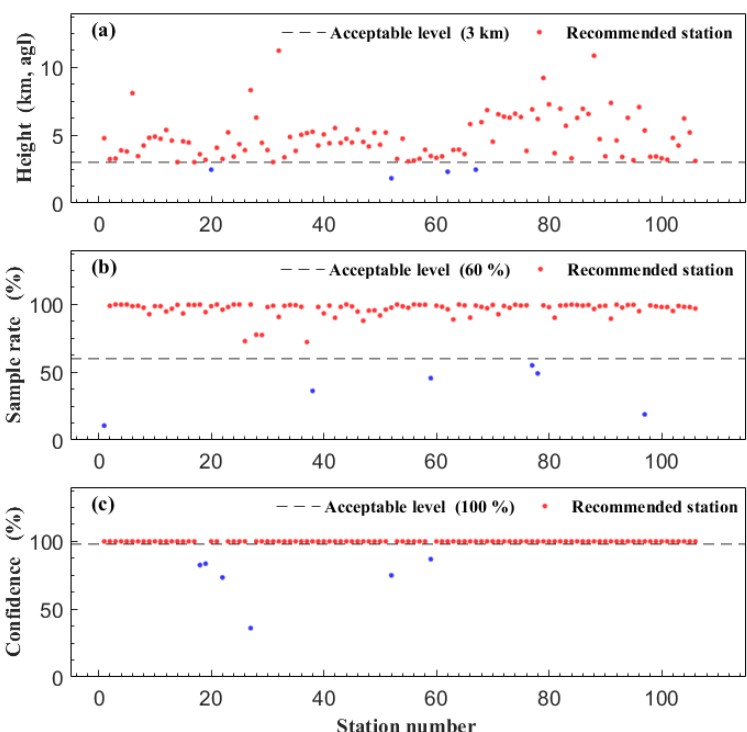

**Figure 5.** Recommended and unrecommended sites of the radar wind profiler Network under different performance index: (a) effective height detected by RWP, (c) data acquisition rate, and (e) data confidence. The horizontal gray lines indicate the corresponding acceptable level.

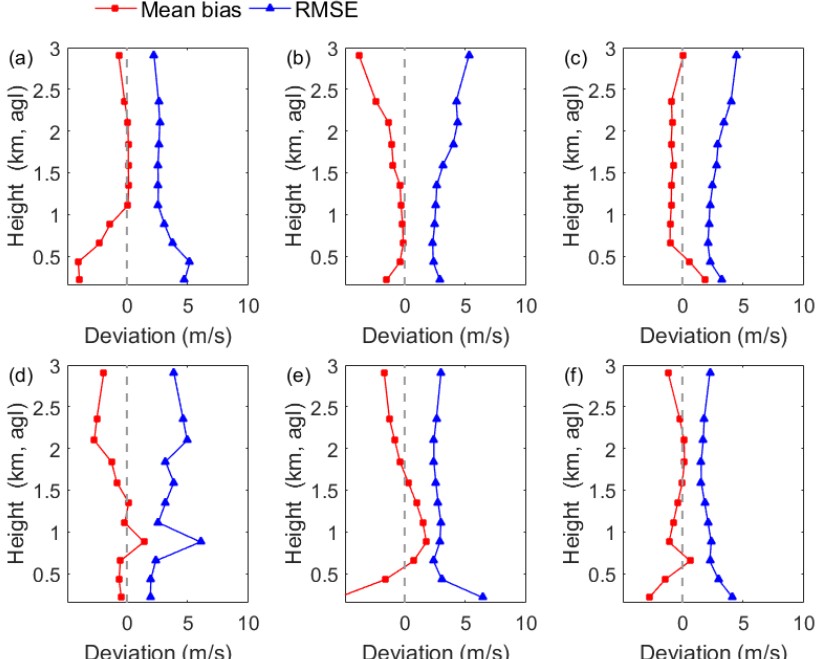

**Figure 6.** Comparison results between RWP and ECMWF at six typical stations: (a) 54511 at Beijing site, (b) 58238 at Nanjing site, (c) 57516 at Chongqing site, (d) 58460 at Shanghai site, (e) 59488 at Zhuhai site, and (f) 59758 at Haikou site. The grey, red and blue dot lines represent the the reference line, mean bias and RMSE, respectively.



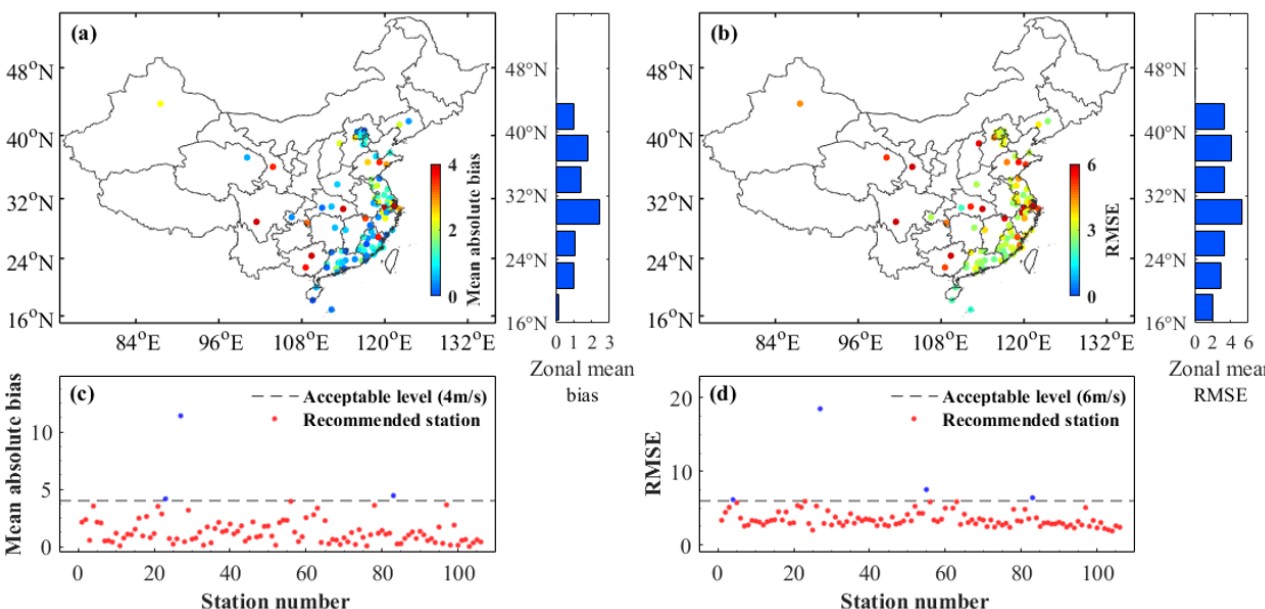

**Figure 7.** Spatial distribution of (a) mean absolute bais and (b) RMSE at each station during November 2018 to March 2019; the corresponding histogram represent the average deviation in zonal direction; (c) and (d) are corresponding recommended and unrecommended sites for (a) and (b), respectively.



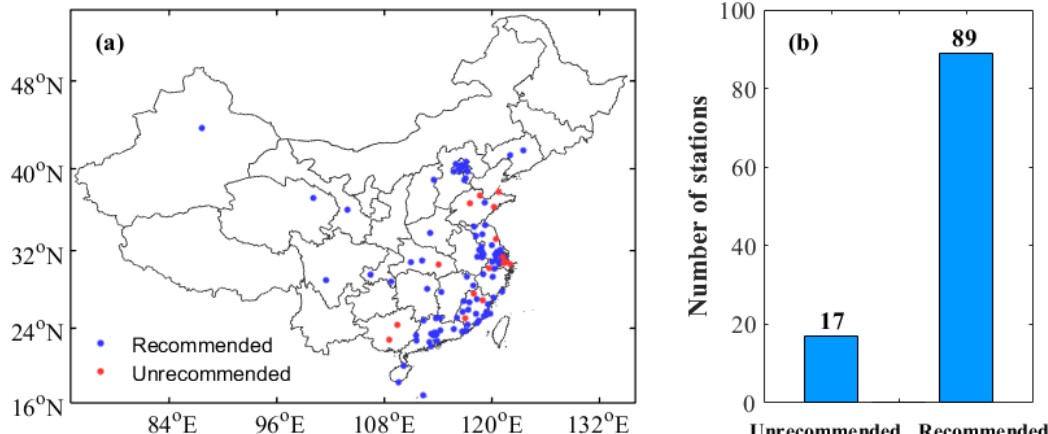

**Figure 8.** (a) Recommended and unrecommended sites of the radar wind profiler network of China. The red and blue dots represent the unrecommended and recommended sites, respectively. (b) shows

5   their corresponding histograms for the statistics regarding the sites shown in (a).

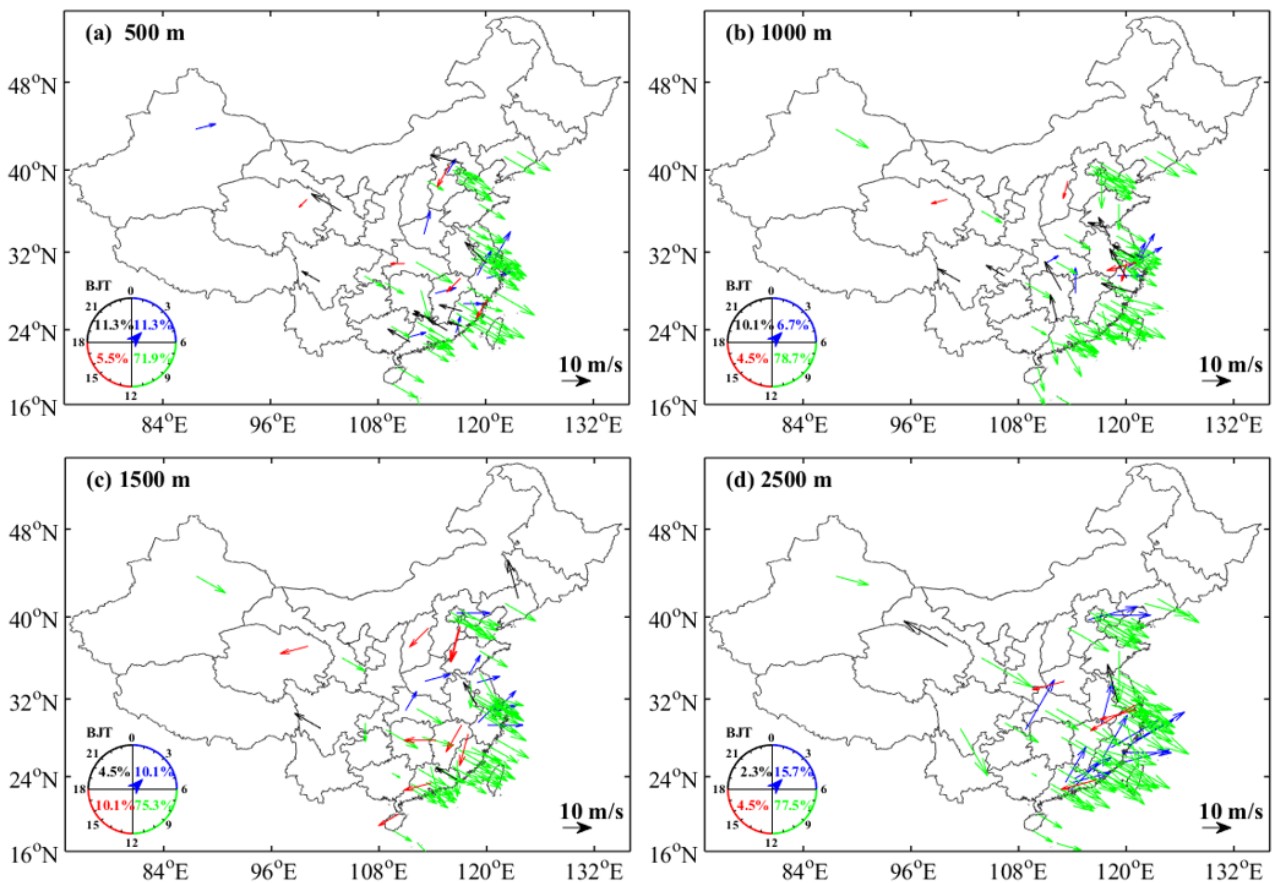

**Figure 9.** Diurnal phase and amplitude of mean maximum wind speed over the period from November 2018 to March 2019 at (a) 500 m, (b) 1000 m, (c) 1500 m, and (d) 2500 m above ground level (AGL). The direction towards which an arrow points denotes the Beijing time (BJT) when the maximum occurs (shown on the clock dial in the bottom left corner of each panel) and the arrow length represents magnitudes of mean maximum wind speed. The arrow color denotes varying diurnal phases: blue (0000-0600 BJT), green (0600-1200 BJT), red (1200-1800 BJT) and black (1800-2400 BJT).



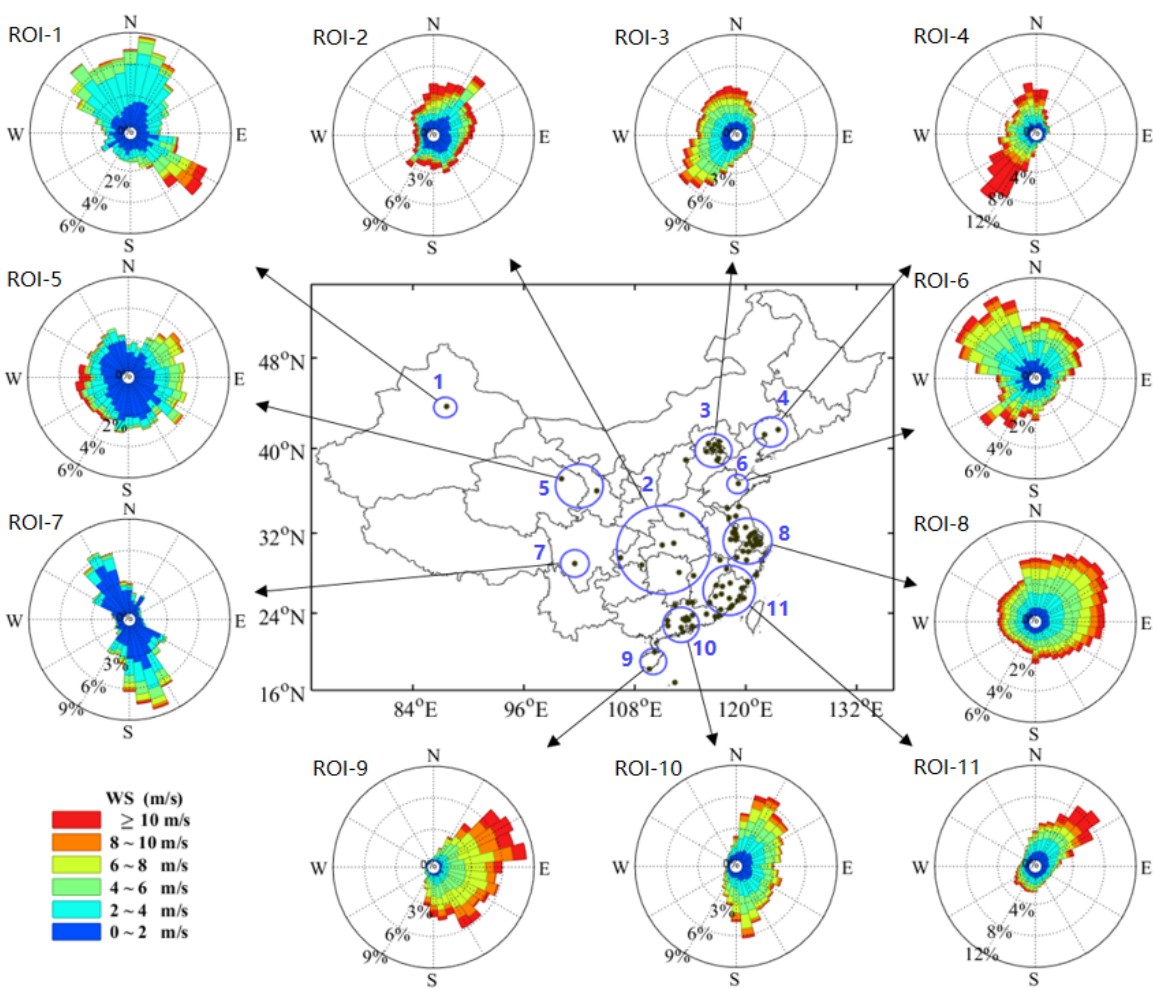

**Figure 10.** Statistical results of atmospheric wind fields at 500 m above ground level (AGL) for 11 region of interest. The wind rose plots in these 11 ROIs are calculated from hourly observations of wind direction and wind speed during November 2018 to March 2019.

