# Peer review of "Characteristics and performance of wind profiles as observed by the radar wind profiler network of China"

_Atmospheric Measurement Techniques, 2020_

## Referee Comment (RC1) · Boming Liu et al. · 25 Mar 2020

radar wind profiler network of China" *by*
Boming Liu et al.

Anonymous Referee #1

This paper examines the characteristics and performance of vertical wind measured by China Radar Wind Profiler Network. The topic is interesting and has climate implications in evaluating and using the wind observations with their reliabilities from Wind Profiler site over China. The paper is well organized and written. The findings of this study are worth of publication in the journal after minor revision. My comment on this paper is mainly related to the evaluation indexes used in the study, which may impact on some of the conclusions. 1. Which criteria are referenced for thresholds of the acceptable levels for these parameters including height, sample rate, confidence (Fig. 5), bias, and RMSE (Fig. 7) in the paper. 2. The ECMWF data set used for validation needs a brief introduction, including its uncertainty and reliability. 3. Note that South

[Figure]

China Sea should be added in the Figures 1,3,4,7,8,910. 4. What does the "ABL" mean in the line 10 of the introduction? Acronyms must be explained in detail when the article first appears. Similar to MODIS at line 8 of page 11 5. The word spacing at P3 L15 need to be modified. 6. For Figure 6, as stations in northwest show much difference with other stations in diurnal phase, any reason? 7. For better cover across the whole China, you should a station in the northwest to make comprehensive comparison with ECMWF. 8. at P10 L18, diurnal phase and amplitude of the mean maximum wind at different heights are almost same. Do authors have any ideas about the reasons that cause this phenomenon? So does the inconsistency for some northwest stations. 9. You used some station names in the text or other figures (e.g., names in figure 6, so should add the locations of these names in figure 1 for readerships. 10. Land use data from MODIS should be descripted in data section. Suggest that MODIS-based land use type should be added as background in Figure 10 for reahderships.

---

## Referee Comment (RC2) · Anonymous Referee #2 · 27 Mar 2020

General

I found the manuscript interesting and reasonably well written, but it needs major revision in my opinion - both to improve clarity and to improve the English. There are also serious questions about the availability outside China of the data described.

Data availability

It is good to know that there is a Chinese wind profiler network. As the authors say on page 4 "the dataset of nationwide profiler network in China has never been revealed". On page 13 they say that the data were provided via http://data.cma.cn/en/. I have looked at that web site and failed to find any mention of wind profilers - more information on data access should be provided. Does one need to register to see wind profiler data? A Chinese colleague looked at the Chinese version and couldn't find wind profilers mentioned there.

The other issue is real-time availability via the Global Telecommunications System (GTS). The abstract recommends the data for global numerical weather prediction - which rather assumes real-time usage. The other wind profiler networks mentioned all provide (or provided) reports via the GTS. As with other comments in this review I expect to see a change in the main manuscript not just a note in the response to reviewers. In this case I want to see a statement as to if and when the data will be available via the GTS. Please contact CMA senior management as appropriate to find out.

The fate of the NPN and profiles from aircraft

The manuscript mentions the NOAA Profiler Network (NPN) and provides three references to it - this is fine. However it should also mention that the NPN largely ceased to operate in 2014 and the last stations closed in 2017. See https://madis.ncep.noaa.gov/madis_npn.shtml As I understand it the profilers were reaching the end of their useful lives and it was decided not to replace them. One factor in their non-replacement was the growing number of profiles (wind and temperature) available from aircraft. On a smaller scale most of the UK profiler network has shut in recent years for similar reasons. I am not saying this to imply that the Chinese profiler network should close, but to provide a rounded picture to readers the closure of the NPN and the availability of aircraft profiles should be mentioned.

Operational monitoring of the Chinese network

The results in this manuscript, whilst useful, seem to be an isolated study. To get the most out of such a network there should be near-real-time monitoring (daily, weekly or monthly) perhaps by comparison to CMA forecast fields. If one of the profilers seems to be performing badly then it should be checked out and perhaps subject to extra

maintenance. Is there any such monitoring and feedback to the network managers? Was there any follow-up about the 17 "unrecommended" stations found by this study?

Manuscript title: The web page gives this as "Characteristics and performance of vertical winds as observed by the radar wind profiler network of China" the manuscript as "A vertical wind profile dataset in China based on the operational radar wind profiler network" - they should be consistent. I have a slight preference for the first ("Characteristics ...") but "vertical winds" (suggests w component) should be replaced by "wind profiles".

Detailed comments

page 1, line 10: 'are the foundation' - 'are a foundation'

1,12: 'JMA, NOAA,EUMETNET, and AGBoM' - unexplained acronyms. Particularly as 'countries' are mentioned earlier in the sentence it seems better to use 'Japan, USA, various European countries and Australia'.

1,13-14 'was presented, which consisted' - 'is presented, consisting' (improved English, like some of the other comments)

1,15 'at various altitudes' add 'but mainly in the boundary layer' or 'about 60% of them have a height range between 3 and 5 km' (my estimate from Figure 5a, the authors could be more precise)

1,22 'unrecommended' should be 'not recommended' here and later in the text

2,4 'Nash et al' - 'Nash and Oakley'

2.1-14 There are a lot of references here, perhaps more than necessary.

2,17 'Nash et al' - 'Nash and Oakley'

2,19 'vertical winds' - 'wind profiles'

2,20 'much' - delete

2,22-25 'Since 2018 ... use.' There have been satellite winds before (from cloud tracking and scatterometers), what is new is that Aeolus provides wind profiles (strictly speaking only line-of-sight winds, $\sim$ only one component). The text needs to be modified a bit.

'its dataset still not released for the public access and use' I understand that CMA is involved in the calibration/validation phase and has access to the Aeolus data. Significant work has been needed on bias correction of the winds, see https://www.ecmwf.int/en/about/media-centre/news/2020/ecmwf-starts-assimilating-aeolus-wind-data (reference this or not as you wish). It is a proof of concept satellite and I don't think I would expect public release of the data at this point.

3,6 'vertical wind over regional scale' - 'regional scale wind fields'

3,6-7 'and to fill the data gap left by field campaign' - delete this. NAWDEX and other field campaigns were never operational

3,20 'Nash et al., 2000' - 'Nash and Oakley, 2001'

3,24 'Government' - delete

3,25 'produced' - 'produces'

4,2 'the vertical' - 'vertical'

4,17 'be' - 'been'

4,21 'reference' - too strong, perhaps 'data source'

5,4... suggestion 'The operational Chinese radar wind profiler network started in 2008 with 5 sites transmitting wind profiles to the headquarters ...' The note about 106 stations in March 2019 should be put in chronological order.

Page 5 general - more information about the various instruments should be given including height range and manufacturer. This might be best presented as a table which

could include the number of instruments and the dates for which they operated.

5,18 'was' - 'is'

5,22 'wind profilerr network: one is raw data and the other product data' - 'wind profiler network: raw data and product data'

6,3 "one's" - delete

6,15 'generated for each observation site'

6,16 suggestion 'A few sites use a low level detection mode with high sampling rate - these provide a vertical resolution of 60 m.'

6,18 'the Fig.' - 'Fig.'

6,19 'selected' - 'were selected'

7,3 'tropospheric type I, tropospheric type II' - what does this mean? better covered in section 2

7,5 'are inhomogeneous'

7,10 'ECMWF wind data' - be more specific. From the mention of the Copernicus web site on page 13 I suspect that ERA5 reanalysis data has been used. This should be clearly stated and a reference given (unfortunately I don't think there is a journal paper yet, but there are short articles in the ECMWF newsletter).

7,11 'verify the accuracy' verify is too strong, perhaps 'estimate'

8,5 "100% confidence" I wish I could be 100% confident about any observation! However if that is how the data are labelled it may be best to stick with this.

Page 8 general: are there any features in common between the 'substandard' sites? Any pattern emerging?

8,24 'inversion algorithm' - 'processing algorithm' better?

8,24 'verify' - 'check'

9,1 'applying these observation data' - 'using these observations'

9,2 'Haseler, 2004' seems an odd reference in this context

9,5 'mean bias' - 'mean speed difference'? Either here or in the figure caption it should be made clear if this is 'RWP-ECMWF' or 'ECMWF-RWP' 'RMSE' - not explained, better to refer to 'RMSD - root-mean-square difference' Neither the profiler or ECMWF/ERA winds are error-free. I assume that it is the vector difference - should be explicitly stated.

9,13 'mean absolute bias' - is this in wind speed? or u-, v- wind components? 'magnitude of the speed bias' would be clearer. 'RMSE' - is this on the vector difference?

9,23 'The large difference ... is mostly related to flaws in the algorithm' What sort of flaws? What is the evidence for this? Is it that bad data has not been screened out? (I think that some wind profilers deteriorate gradually over time, and regular maintenance, replacement of aging components is important. I'm not sure if this is as true of boundary layer profiles.)

10,1 'performance can also be compared to the corresponding radiosonde' Has this been done? How well do the results match those vs ECMWF?

10,18 'mean maximum wind speed within the 24 h' Is this the maximum of the hourly speeds or the half-hourly speeds mentioned earlier? In either case I presume that it is an average over the whole 60 or 30 minutes. (Or perhaps 6-minute winds, Fig 2.) Please clarify.

10,22 'morning ... early morning' 6-12 and 0-6 Beijing time. Clarify these and also 'afternoon', 'evening'

10,15 'Extreme weather detection' - 'Daily maximum winds' better Extreme weather usually means very rare events (say once a year) not daily maxima.

10,16 'extreme weather' - 'diurnal cycle'

10,23 'have an afternoon peak', 'have an evening peak' (not 'the')

11,2 'heights'

11,4 'is inconsistent' (sounds as if they are wrong), 'show a different pattern' better

11,5 'twice' - 'two'

11,5 'indicative that' (not 'of')

11,12 'were selected'

11,19 'Western China' (delete 'The')

11,21 'western China' (delete 'the')

11,23 'both the' - delete

11,25 'the 30%' - delete 'the'; 'over the whole study period'

12,1 'measurements'

12,3 'wind turbines' - perhaps mention that winds from turbine height (60-100 m?) would be useful for NWP (more useful than a 10 m wind from Synop stations)

12,9 'Wind profiles are of great ...'

12,13 'consisted' - 'consists'

12,18 'mean absolute bias' - see note on page 9, also state what is being used in the comparison.

12,21 'extreme weather'? I expect they could be used for this, but this hasn't been shown, see discussion on page 10.

13,20-21 two closing brackets ")" without opening ones "(".

Table 1. The caption should mention Figure 10, which is the closest to a definition of the ROIs. Perhaps a table of all stations, (identifier, latitude, longitude, altitude, ROI, type of profiler, land cover, quality flags) could be provided as an appendix or supplemental information. Column 2 of table 1 should I think be 'Number of sites'.

Figure 3. This could be quite interesting, but the panels are too small for readers to see clearly. My suggestion would be just to show four levels as in Figure 9 (probably the same four levels) and not bother with the stacked plot e).

Figure 4 caption 'detection'

Figure 6 'mean bias' - 'mean speed difference'?

Figure 7 'mean absolute bias' (whatever that means) and 'RMSE'. In this figure the values seem to have been averaged over levels (all levels, or levels up to 3 km?), clarification is needed. See discussion on page 9.

Figure 8. The histogram 8b) is superfluous, I would remove it. Suggestion for caption: 'The blue dots represent the 89 recommended sites and red dots the 17 non-recommended sites.'

---

## Author Comment (AC1) · 20 May 2020

**Response to Reviewer 2's Comments**

***Response: We thank the anonymous reviewer for his/her comprehensive evaluation and thoughtful comments. We have addressed the reviewer's concern one by one. For clarity purpose, here we have listed the reviewers' comments in plain font, followed by our response in bold italics***

I found the manuscript interesting and reasonably well written, but it needs major revision in my opinion - both to improve clarity and to improve the English. There are also serious questions about the availability outside China of the data described.

***Response: Thanks for your critical but valuable comments on our manuscript, which helps great in improving the quality of our manuscript. Please see the following point-by-point response to your comments.***

1: It is good to know that there is a Chinese wind profiler network. As the authors say on page 4 "the dataset of nationwide profiler network in China has never been revealed". On page 13 they say that the data were provided via http://data.cma.cn/en/. I have looked at that web site and failed to find any mention of wind profilers - more information on data access should be provided. Does one need to register to see wind profiler data? A Chinese colleague looked at the Chinese version and couldn't find wind profilers mentioned there.

***Response: Good question. In consultation with the director of CMA data management department, we got to know that the RWP network is still under construction, and the data quality has to be assessed before release. Therefore, this motivates us make preliminary assessment of this dataset in this study. As long as the quality meets the national standard or numerical weather prediction, all of these measurements will be available to the public upon request through the website (*** http://data.cma.cn/en/ ***). Nevertheless, it requires the user to register an account and submit an application form in order to get access. Currently, parts of the data are***

*available for research. Therefore, in this revision, we modified the description of data availability. "The radar wind profiler data used in this paper can be provided for non-commercial research purposes upon request by email (Dr. Jianping Guo: jpguocams@gmail.com)."*

2: The other issue is real-time availability via the Global Telecommunications System (GTS). The abstract recommends the data for global numerical weather prediction - which rather assumes real-time usage. The other wind profiler networks mentioned all provide (or provided) reports via the GTS. As with other comments in this review I expect to see a change in the main manuscript not just a note in the response to reviewers. In this case I want to see a statement as to if and when the data will be available via the GTS. Please contact CMA senior management as appropriate to find out.

*Response: First of all, the other wind profiler network mentioned in our manuscript is operational, but this is not the case for the RWP network of China. This is one of the motivations for our study. In consultation with the several CMA senior officials, the RWP network data have to be further assessed before it becomes fully operational. As of 2017, it began to send the data to CMA headquarter. It will be shared via GTS in the next several years, which highly depends on the process of data quality assessment as well as data assimilation. The sharing of this dataset is expected to contribute to the numerical weather prediction and boundary layer meteorology community.*

*To clarify this issue, we added the following statement to section 2: "Due to the fact that the measurements from the RWP network of China have to be further assessed, the data sharing via global telecommunications system is expected to occur in the next several years, which highly depends on the process of data quality assessment."*

3: The fate of the NPN and profiles from aircraft

The manuscript mentions the NOAA Profiler Network (NPN) and provides three references to it - this is fine. However, it should also mention that the NPN largely

ceased to operate in 2014 and the last stations closed in 2017. See https://madis.ncep.noaa.gov/madis_npn.shtml. As I understand it the profilers were reaching the end of their useful lives and it was decided not to replace them. One factor in their non-replacement was the growing number of profiles (wind and temperature) available from aircraft. On a smaller scale most of the UK profiler network has shut in recent years for similar reasons. I am not saying this to imply that the Chinese profiler network should close, but to provide a rounded picture to readers the closure of the NPN and the availability of aircraft profiles should be mentioned.

*Response: Good point! The information you provided is of great importance. Per your kind suggestion, we added some descriptions mentioning the closure of the NPN and the availability of aircraft profiles in the introduction section, which shows as follows:*

*"Nevertheless, probably due to the fact that the RWP reached the end of their useful lives, the NPN largely ceased to operate in 2014 and the last stations closed in 2017. As an alternative data source, the high-density airborne wind and temperature profiles from civil aviation industry gradually took over the role of RWP since then (https://madis.ncep.noaa.gov/madis_npn.shtml)."*

4: Operational monitoring of the Chinese network

The results in this manuscript, whilst useful, seem to be an isolated study. To get the most out of such a network there should be near-real-time monitoring (daily, weekly or monthly) perhaps by comparison to CMA forecast fields. If one of the profilers seems to be performing badly then it should be checked out and perhaps subject to extra maintenance. Is there any such monitoring and feedback to the network managers? Was there any follow-up about the 17 "unrecommended" stations found by this study?

*Response: This network is currently not operational, and this work is an initial attempt to assess the data quality of RWP network of China. More importantly, another purpose of our manuscript is to let the atmospheric science community know the existence of such observational network. In the long run, the data will be disseminated to the public.*

*Regarding how to get the most out of this network, we totally agree with you that we should compare the near-real-time monitoring at daily, weekly or monthly timescales against the CMA forecast fields, which is the focus of our future work. Thanks for your kind suggestions.*

*In the meanwhile, the CMA officials have already known the preliminary assessment results of this network, since the coauthor of Dr. Lijuan Shi, one of the persons in charge of RWP operation and maintenance. reported the status of RWP network to her boss. The double-check and careful maintenance of the RWP instruments are undergoing at the 17 "unrecommended" stations.*

5: Manuscript title: The web page gives this as "Characteristics and performance of vertical winds as observed by the radar wind profiler network of China" the manuscript as "A vertical wind profile dataset in China based on the operational radar wind profiler network" - they should be consistent. I have a slight preference for the first ("Characteristics ...") but "vertical winds" (suggests w component) should be replaced by "wind profiles".

*Response: Good suggestion! The title has been modified to "Characteristics and performance of wind profiles as observed by the radar wind profiler network of China"*

6: page 1, line 10: 'are the foundation' - 'are a foundation'

*Response: It has been revised to "fundamental to"*

7: 1,12: 'JMA, NOAA, EUMETNET, and AGBoM' - unexplained acronyms. Particularly as "countries" are mentioned earlier in the sentence it seems better to use 'Japan, USA, various European countries and Australia'.

*Response: Amended as suggested.*

8: 1,13-14 'was presented, which consisted' - 'is presented, consisting' (improved English, like some of the other comments)

*Response: Amended as suggested.*

9: 1,15 'at various altitudes add 'but mainly in the boundary layer' or 'about 60% of them have a height range between 3 and 5 km' (my estimate from Figure 5a, the authors could be more precise)

*Response: We have added 'but mainly in the boundary layer' after 'at various altitudes"*

10: 1,22 'unrecommended' should be 'not recommended' here and later in the text.

*Response: Amended as suggested.*

11: 2,4 'Nash et al' - 'Nash and Oakley'

*Response: Amended as suggested.*

12: 2.1-14 There are a lot of references here, perhaps more than necessary.

*Response: We have deleted some references.*

13: 2,17 'Nash et al' - 'Nash and Oakley'

*Response: This reference has been deleted.*

14: 2,19 'vertical winds' - 'wind profiles'

*Response: Amended as suggested.*

15: 2,20 'much' – delete

*Response: Amended as suggested.*

16: 2,22-25 'Since 2018 ... use.' There have been satellite winds before (from cloud tracking and scatterometers), what is new is that Aeolus provides wind profiles (strictly speaking only line-of-sight winds, ∼ only one component). The text needs to be modified a bit.

"its dataset still not released for the public access and use" I understand that CMA is involved in the calibration/validation phase and has access to the Aeolus data. Significant work has been needed on bias correction of the winds, see https://www.ecmwf.int/en/about/media-centre/news/2020/ecmwf-startsassimilating-aeolus-wind-data (reference this or not as you wish). It is a proof of concept satellite and I don't think I would expect public release of the data at this point.

*Response: We appreciate your insightful comments. Given the fact that the Aeolus data has been released on May 12, 2020, in the introduction section we rewrote the related description as follows:*

*"The earliest space-borne wind products generally refer to the atmospheric motion vectors that are derived by tracking clouds or areas of water vapor through consecutive infrared remote sensing images (Schmetz et al., 1993; Velden et al., 2005). Later on, the vector winds over the ocean surface have been measured by the spaceborne microwave instruments such as SeaWinds onboard QuikSCAT (Bentamy et al., 1999; Draper and Long 2002). Since 2018, new satellite-based wind observational era set in with the launch of European space agency (ESA)'s Aeolus wind satellite on which the direct-detection Doppler wind lidar ALADIN is accommodated, which provides line-of-sight winds along the satellite track (Reitebuch et al., 2009; Reitebuch 2012). As of 12 May 2020, the Aeolus data has gone public after the bias correction of the winds has been adequately made, which are now being distributed publicly to forecasting services and scientific users in less than three hours of measurements being made from space (https://www.esa.int/Applications/Observing_the_Earth/Aeolus/Aeolus_goes_public )."*

17: 3,6 'vertical wind over regional scale' - 'regional scale wind fields'

*Response: Amended as suggested.*

18: 3,6-7 'and to fill the data gap left by field campaign' - delete this. NAWDEX and other field campaigns were never operational

*Response: Amended as suggested.*

19: 3,20 'Nash et al., 2000' - 'Nash and Oakley, 2001'

*Response: Amended as suggested.*

20: 3,24 'Government' – delete

*Response: Amended as suggested.*

21: 3,25 'produced' - 'produces'

*Response: Amended as suggested.*

22: 4,2 'the vertical' - 'vertical'

*Response: Amended as suggested.*

23: 4,17 'be' - 'been'

*Response: Amended as suggested.*

24: 4,21 'reference' - too strong, perhaps 'data source'

*Response: Amended as suggested.*

25: 5,4... suggestion 'The operational Chinese radar wind profiler network started in 2008 with 5 sites transmitting wind profiles to the headquarters ...' The note about 106 stations in March 2019 should be put in chronological order.

*Response: Amended as suggested.*

26: Page 5 general - more information about the various instruments should be given including height range and manufacturer. This might be best presented as a table which could include the number of instruments and the dates for which they operated.

***Response: Per your suggestion. We add a table detailing the various instruments in this section.***

*Table 1. Instrument information of radar wind profiler network of China*

| Type of RWP | Identifier | Max detection height | Frequency | # of sites | Manufacturer |
|---|---|---|---|---|---|
| High Troposphere (CFL-16) | PA | 8-10 km | 440-450 MHz | 3 | CASIC |
| Low Troposphere (CFL-08) | PB | 6-8 km | 440-450 MHz | 2 | CASIC |
| Boundary layer | LC | 3-5 km | 1290 MHz | 101 | CASIC/CETC/CHG |

CASIC: China Aerospace Science & Industry Corp.

CETC: China Electronics Technology Group Corp.

CHG: China Huayun Meteorological Technology Group Corp.

27: 5,18 'was' - 'is'

***Response: Amended as suggested.***

28: 5,22 'wind profilerr network: one is raw data and the other product data' - 'wind profiler network: raw data and product data'

***Response: Amended as suggested.***

29: 6,3 "one's" – delete

***Response: Amended as suggested.***

30: 6,15 'generated for each observation site'

***Response: Amended as suggested.***

31: 6,16 suggestion 'A few sites use a low level detection mode with high sampling rate -these provide a vertical resolution of 60 m.'

*Response: Amended as suggested.*

32: 6,18 'the Fig.' - 'Fig.'

*Response: Amended as suggested.*

33: 6,19 'selected' - 'were selected'

*Response: Amended as suggested.*

34: 7,3 'tropospheric type I, tropospheric type II' - what does this mean? better covered in section 2

*Response: "tropospheric type I and tropospheric type II" indicates high troposphere, low troposphere RWP, respectively. To avoid misunderstanding, we have revised the name of the types of RWP, which is detailed in the newly-added Table 1. The corresponding descriptions in section 2 are as follows:*

*"Table 1 shows the instrument information of RWP used in this study, which consist of three types of RWP: high troposphere, low troposphere and boundary layer RWPs. It can be seen that the majority of the radars are boundary layer RWP operating at L band (101 sites), and a few of sites are instrumented with tropospheric RWP operating at P band (5 sites)."*

35: 7,5 'are inhomogeneous'

*Response: Amended as suggested.*

36: 7,10 'ECMWF wind data' - be more specific. From the mention of the Copernicus web site on page 13 I suspect that ERA5 reanalysis data has been used. This should be clearly stated and a reference given (unfortunately I don't think there is a journal paper yet, but there are short articles in the ECMWF newsletter).

*Response: Per your suggestion, we rewrote this sentence as follows:*

*"In order to estimate the data accuracy, the wind profiles from RWP are compared with hourly wind measurements at 0.25 x 0.25-degree latitude/longitude grid from*

*the fifth generation European Centre for Medium-range Weather Forecasts (ECMWF) atmospheric reanalysis of the global climate (ERA5, Hoffmann et al., 2019).*

37: 7,11 'verify the accuracy' verify is too strong, perhaps 'estimate'

*Response: Amended as suggested.*

38: 8,5 "100% confidence" I wish I could be 100% confident about any observation! However, if that is how the data are labelled it may be best to stick with this.

*Response: Agreed. The "confidence" is the quality flag of data. As indicated in the user manual of RWP, only those measurements with a 100% confidence level can be fully trusted. However, as your said, it is almost impossible for all observations are guaranteed to be 100% confident. Therefore, the data assessment is made based on the criterion of 100% confidence.*

39: Page 8 general: are there any features in common between the 'substandard' sites? Any pattern emerging?

*Response: There are a total of 14 substandard sites, and the emerging patterns for the substandard sites differ greatly, which means that not any common features were revealed for these sites.*

40: 8,24 'inversion algorithm' - 'processing algorithm' better?

*Response: Amended as suggested.*

41: 8,24 'verify' - 'check'

*Response: Amended as suggested.*

42: 9,1 'applying these observation data' - 'using these observations'

*Response: Amended as suggested.*

43: 9,2 'Haseler, 2004' seems an odd reference in this context

*Response: We have deleted this reference.*

44: 9,5 'mean bias' - 'mean speed difference'? Either here or in the figure caption it should be made clear if this is 'RWP-ECMWF' or 'ECMWF-RWP' 'RMSE' - not explained, better to refer to 'RMSD - root-mean-square difference' Neither the profiler or ECMWF/ERA winds are error-free. I assume that it is the vector difference - should be explicitly stated.

*Response: Good suggestions. Per your kind suggestion, the "mean bias" was revised to "mean speed difference", which indicated the mean wind speed difference between RWP and ECMWF (RWP–ECMWF) at each height. Likewise, the "RMSE" was revised to "RMSD" (root-mean-square difference). All these suggestions have been taken in this revision.*

45: 9,13 'mean absolute bias' - is this in wind speed? or u-, v- wind components? 'magnitude of the speed bias' would be clearer. 'RMSE' - is this on the vector difference?

*Response: Per your suggestion, the sentence has been revised as follows:*
*"Here, the horizontal wind speed measurements at all levels ranging from 0 to 3 km are used to calculate the MSD and RMSD at each site. Moreover, the magnitude of mean speed difference (MMSD) and RMSD are set to be 4 m/s and 6 m/s, respectively, which serve as a target for acceptable criterion"*

46: 9,23 'The large difference ... is mostly related to flaws in the algorithm' What sort of flaws? What is the evidence for this? Is it that bad data has not been screened out? (I think that some wind profilers deteriorate gradually over time, and regular maintenance, replacement of aging components is important. I'm not sure if this is as true of boundary layer profiles.)

*Response: Per your suggestion. We modified this sentence. "The large difference may be caused by hardware or configuration problems, such as the aging of components. Therefore, it is important to conduct regular maintenance and replacement of aged components."*

47: 10,1 'performance can also be compared to the corresponding radiosonde' Has this been done? How well do the results match those vs ECMWF?

*Response: This sentence has been deleted, since we did not do the comparison analysis.*

48: 10,18 'mean maximum wind speed within the 24 h' Is this the maximum of the hourly speeds or the half-hourly speeds mentioned earlier? In either case I presume that it is an average over the whole 60 or 30 minutes. (Or perhaps 6-minute winds, Fig 2.) Please clarify.

*Response: Amended as suggested. The "mean maximum wind speed" was replaced to "mean maximum hourly wind speed".*

49: 10,22 'morning ... early morning' 6-12 and 0-6 Beijing time. Clarify these and also' afternoon', 'evening'

*Response: It has been clarified as follows:*

*"The occurrence time of maximum hourly wind speed is marked as early morning (0000–0600 Beijing time, BJT, ), morning (0600–1200 BJT), afternoon (1200–1800 BJT), and evening (1800–2400 BJT), respectively.*

50: 10,15 'Extreme weather detection' - 'Daily maximum winds' better Extreme weather usually means very rare events (say once a year) not daily maxima.

*Response: Amended as suggested.*

51: 10,16 'extreme weather' - 'diurnal cycle'

*Response: Amended as suggested.*

52: 10,23 'have an afternoon peak', 'have an evening peak' (not 'the')

*Response: Amended as suggested.*

53: 11,2 'heights'

*Response: Amended as suggested.*

54: 11,4 'is inconsistent' (sounds as if they are wrong), 'show a different pattern' better

*Response: Amended as suggested.*

55: 11,5 'twice' - 'two'

*Response: Amended as suggested.*

56: 11,5 'indicative that' (not 'of')

*Response: Amended as suggested.*

57: 11,12 'were selected'

*Response: Amended as suggested.*

58: 11,19 'Western China' (delete 'The')

*Response: Amended as suggested.*

59: 11,21 'western China' (delete 'the')

*Response: Amended as suggested.*

60: 11,23 'both the' – delete

*Response: Amended as suggested.*

61: 11,25 'the 30%' - delete 'the'; 'over the whole study period'

*Response: Amended as suggested.*

62: 12,1 'measurements'

*Response: Amended as suggested.*

63: 12,3 'wind turbines' - perhaps mention that winds from turbine height (60-100 m?) would be useful for NWP (more useful than a 10 m wind from Synop stations)

*Response: Amended as suggested. "The policymakers will determine whether the wind turbines (60-100 m above ground level) will be installed or not, aided by high-resolution model simulation analyses."*

64: 12,9 'Wind profiles are of great ...'

*Response: Amended as suggested.*

65: 12,13 'consisted' - 'consists'

*Response: Amended as suggested.*

66: 12,18 'mean absolute bias' - see note on page 9, also state what is being used in the comparison.

*Response: Per your suggestion. The "mean absolute bias" was revised to "magnitude of mean speed difference (MMSD)". The MMSD at each site were calculated from the horizontal wind speed at all levels from 0 to 3 km between RWP and ECMWF.*

67: 12,21 'extreme weather'? I expect they could be used for this, but this hasn't been shown, see discussion on page 10.

*Response: Per your suggestion. The "extreme weather" was revised to "daily maximum winds".*

68: 13,20-21 two closing brackets ")" without opening ones "(".

*Response: Amended as suggested.*

69: Table 1. The caption should mention Figure 10, which is the closest to a definition of the ROIs. Perhaps a table of all stations, (identifier, latitude, longitude, altitude, ROI, type of profiler, land cover, quality flags) could be provided as an appendix or supplemental information. Column 2 of table 1 should I think be 'Number of sites'.

*Response: Per your suggestion, we modified the Table, as shown below:*

***Table 2.*** *Statistics of the number of sites and land cover types for the 11 regions of interest (ROI) in Fig. 10.*

| Region of interest | Number of sites | Land cover types |
|---|---|---|
| 1 | 1 | Grassland |
| 2 | 7 | Cropland and Forest |
| 3 | 10 | Urban |
| 4 | 2 | Cropland |
| 5 | 2 | Grassland |
| 6 | 1 | Cropland |
| 7 | 1 | Grassland |
| 8 | 27 | Urban |
| 9 | 2 | Cropland and Forest |
| 10 | 10 | Urban |
| 11 | 19 | Urban and Forest |

*In addition, we added a table (Table S1) showing all stations (identifier, latitude, longitude, altitude, type of profiler, land cover) in the supplemental materials.*

70: Figure 3. This could be quite interesting, but the panels are too small for readers to see clearly. My suggestion would be just to show four levels as in Figure 9 (probably the same four levels) and not bother with the stacked plot e).

*Response: Per your suggestion. The Fig.3 was modified as following:*

[Figure]

71: Figure 4 caption 'detection'

*Response: This typo has been corrected as suggested.*

72: Figure 6 'mean bias' - 'mean speed difference'?

*Response: Amended as suggested.*

73: Figure 7 'mean absolute bias' (whatever that means) and 'RMSE'. In this figure the values seem to have been averaged over levels (all levels, or levels up to 3 km?), clarification is needed. See discussion on page 9.

*Response: You are right, and these values were calculated over all the levels up to 3 km. Per your suggestion. Therefore, the modified caption of Fig.7 now reads as follows: "Spatial distribution of (a) magnitude of the mean speed difference (MMSD) and (b) root-mean-square difference (RMSD) at each station during November 2018 to March 2019; the corresponding histogram represent the average difference in zonal direction; (c) and (d) are corresponding recommended (red dots) and non-*

*recommended (blue dots) sites for (a) and (b), respectively. The MMSD and REMD at each station were derived from the measurements over all levels from 0–3 km."*

74: Figure 8. The histogram 8b) is superfluous, I would remove it. Suggestion for caption: 'The blue dots represent the 89 recommended sites and red dots the 17 nonrecommended sites.'

*Response: Amended as suggested.*

---

## Author Comment (AC2) · 20 May 2020

**Response to Reviewer 1's Comments**

*Response: We thank the anonymous reviewer for his/her comprehensive evaluation and thoughtful comments. We have addressed the reviewers' concern one by one. For clarity purpose, here we have listed the reviewer' comments in plain font, followed by our response in bold italics.*

This paper examines the characteristics and performance of vertical wind measured by China Radar Wind Profiler Network. The topic is interesting and has climate implications in evaluating and using the wind observations with their reliabilities from Wind Profiler site over China. The paper is well organized and written. The findings of this study are worth of publication in the journal after minor revision. My comment on this paper is mainly related to the evaluation indexes used in the study, which may impact on some of the conclusions.

*Response: Thanks for the reviewer's positive comments on our manuscript.*

1. Which criteria are referenced for thresholds of the acceptable levels for these parameters including height, sample rate, confidence (Fig. 5), bias, and RMSE (Fig. 7) in the paper.

*Response: Good question. In fact, no well-established criteria can be found in the current references for the setting of the parameters you mentioned. Generally, the criteria are formulated according to the needs of users. We hope to apply the radar wind profiler network data to continuous atmospheric boundary layer observations in the subsequent research. Therefore, it required that the effective detection height above 3 km, the acquisition rate above 60%, the confidence level reach 100%, the mean speed difference less than 4 m/s, and mean RMSD less than 6 m/s.*

*To clarify this issue, we thoroughly revised the following paragraph in section 3.1: "In order to make the criteria of RWP network data consistent, we have to set corresponding screening criteria for each system index, which to some degrees reflects the needs of future applications. For instance, the RWP network data are*

*expect to be used to derive boundary layer parameters, such as boundary layer height (Liu et al., 2019) and wind shear that are closely related to atmospheric pollution (Zhang et al., 2020). Therefore, it would be better for the effective detection height of RWP reaching 3 km, with the acquisition rate being above 60%. In addition, according to the user manual of RWP, only those wind profile data with a 100% confidence level are recommended."*

*Besides, in section 3.2, we rewrote the following paragraph:*

*"Here, the horizontal wind speed measurements at all levels ranging from 0 to 3 km are used to calculate the MSD and RMSD at each site. Moreover, the magnitude of mean speed difference (MMSD) and RMSD are set to be 4 m/s and 6 m/s, respectively, which serve as a target for acceptable criterion"*

2. The ECMWF data set used for validation needs a brief introduction, including its uncertainty and reliability.

*Response: Per your suggestion. By referring to previous studies on the assessment of ERA5 reanalysis, we added the following sentences in section 3. "In order to estimate the data accuracy, the wind profiles from RWP are compared with hourly wind measurements at 0.25 x 0.25-degree latitude/longitude grid from the fifth generation European Centre for Medium-range Weather Forecasts (ECMWF) atmospheric reanalysis of the global climate (ERA5, Hoffmann et al., 2019).*

3. Note that South China Sea should be added in the Figures 1,3,4,7,8,910.

*Response: Amended as suggested.*

4. What does the "ABL" mean in the line 10 of the introduction? Acronyms must be explained in detail when the article first appears. Similar to MODIS at line 8 of page 11.

*Response: The "ABL" refers to "planetary boundary layer (PBL)", which has been corrected to make it consistent throughout this manuscript. The "MODIS" is short for "the Moderate-resolution Imaging Spectroradiometer".*

5. The word spacing at P3 L15 need to be modified.

*Response: Amended as suggested.*

6. For Figure 6, as stations in northwest show much difference with other stations in diurnal phase, any reason?

*Response: If I understand correctly, you should refer to Fig. 9. We think these differences may be caused by differences in climate and terrain. Northwest China is a semi-arid and plateau region, but most other sites are distributed in the plains.*

7. For better cover across the whole China, you should a station in the northwest to make comprehensive comparison with ECMWF.

*Response: Per your kind suggestion, the Nanjing site in the eastern China was replaced Wulumuqi, one site located in northwestern China. Also, the Zhuhai site was replaced with Zigui.*

8. at P10 L18, diurnal phase and amplitude of the mean maximum wind at different heights are almost same. Do authors have any ideas about the reasons that cause this phenomenon? So does the inconsistency for some northwest stations.

*Response: Here, we present the statistical results of the diurnal phase and amplitude of the mean maximum wind at different heights. Due to the atmospheric wind field is affected by multiple factors such as vertical pressure, temperature gradient, terrain and climate. Therefore, it is hard to clarify the specific reason. We think it may be due to the downhill mountain winds or offshore winds.*

9. You used some station names in the text or other figures (e.g., names in figure 6), so should add the locations of these names in figure 1 for readerships.

*Response: Due to there are many observation sties in Beijing and Shanghai area, it is hard to add station name in the Fig.1. Therefore, we add the latitude and longitude of the six stations in text for readers.*

10. Land use data from MODIS should be described in data section. Suggest that MODIS-based land use type should be added as background in Figure 10 for readerships.

*Response: Per your suggestion, the land cover data and their description has been added in section 4.2 of this revision. Now it reads as follows:*

*"The MODIS Land Cover product is derived through a supervised decision-tree classification method. The land cover types are divided into 17 classes, including 11 natural vegetation classes, three human-altered classes, and three non-vegetated classes (Friedl et al., 2019)."*

*Moreover, the land cover data was added as background in the Fig.10.*

[Figure]

***Figure 10. Spatial distribution of the statistical results of atmospheric wind fields at 500 m above ground level (AGL) for 11 regions of interest (ROIs). The wind rose plots over the 11 ROIs are calculated from hourly observations of wind direction and wind speed from November 2018 to March 2019. The land cover types 0–16 represent the Water, Evergreen Needleleaf forest, Evergreen Broadleaf forest, Deciduous Needleleaf forest, Deciduous Broadleaf forest, Mixed forest, Closed shrublands, Open shrublands, woody savannas, Savannas, Grasslands, Permanent Croplands,***

*Urban and built-up, Cropland/Natural vegetation mosaic, Snow and ice, Barren or sparsely vegetated, respectively.*

---

## Author Response (AR2)

**Author's Response**

**Associate Editor Comments to the Author:**

Where the first review round changes are much appreciated, the second reviewer brings up some pertinent points for your consideration and for clarification. Looking forward to your further

5  improvements.

*Response: We thank the anonymous reviewers for their comprehensive evaluation and thoughtful comments. We have addressed the reviewer's concern one by one. For clarity purpose, here we have listed the reviewers' comments in plain font, followed by our response in bold italics*

10  **Anonymous Referee #2:**

The manuscript has been improved from the first version and should be acceptable after minor changes.
*Response: Thank you very much for your positive comments.*

15  There are two very recent publications that should be useful to the authors (full details below). Hersbach et al (2020, QJRMS - Early View) describes ERA5 and should certainly be referenced. In the text most references to ECMWF should be to ERA5. Rennie and Isaksen (2020, ECMWF TM 864) describes the impact of Aeolus winds in the operational ECMWF NWP system.

*Response: Per your suggestion, the two recent papers you suggested have been cited in this*

20  *revision.*

13, 9 'vertical wind speed' Yes wind profilers can measure the vertical component of the wind, but no results have been presented for it and it isn't clear how useful it is. Either omit 'vertical wind speed' or mention these caveats.

*Response: "vertical wind speed" was deleted as suggested.*

16, 19 'Haseler ...' No longer referenced, omit

*Response: Amended as suggested.*

Supplement - I was pleased to see this table added.
I suggest that the latitude and longitude should be given to two or three decimal places.
The first ID '16078' stands out as different - is this correct?

*Response: We agree that the data sharing of the exact location of RWP stations in the scientific community is significant and crucial. However, I am afraid that this data cannot be made publicly available by any Chinese citizens, because these data are extremely sensitive according to the Meteorology Law of the People's Republic of China. Therefore, we can only provide the approximate location of the RWP sites.*

*Of course, under the premise of non-commercial scientific research activities, small-scale data sharing is possible. Therefore, we claim in the section of "Data availability" as follows:*
*"The radar wind profiler data used in this paper can be provided for non-commercial research purposes upon reasonable request (Dr. Jianping Guo, Email: jpguocams@gmail.com)."*

*This statement is also in accordance with the data policy of AMT Journal ([https://www.atmospheric-measurement-techniques.net/about/data_policy.html](https://www.atmospheric-measurement-techniques.net/about/data_policy.html)). It reads on the*

*website: "If the data are not publicly accessible, a detailed explanation of why this is the case is required."*

     *Concerning the first ID "16078", we checked the site information, and found that the "16078" was the site ID during the test period, and the site ID now is changed to "58933" during the operation. We modified this error in Table S1. Thanks again.*

[revised manuscript text omitted]